# V_H_-Based Mini Q-Body: A Novel Quench-Based Immunosensor

**DOI:** 10.3390/s23042251

**Published:** 2023-02-17

**Authors:** Jinhua Dong, Bhagat Banwait, Hiroshi Ueda, Peter Kristensen

**Affiliations:** 1School of Rehabilitation Sciences and Engineering, University of Health and Rehabilitation Sciences, Qingdao 266071, China; 2Laboratory for Chemistry and Life Science, Institute of Innovative Research, Tokyo Institute of Technology, Yokohama 226-8503, Japan; 3International Research Frontiers Initiative, Tokyo Institute of Technology, Yokohama 226-8503, Japan; 4Department of Engineering, Aarhus University, 8000 Aarhus, Denmark; 5Department of Chemistry and Bioscience, Aalborg University, 9220 Aalborg, Denmark

**Keywords:** Quenchbody, fluorescence sensor, single-domain antibody, hen egg lysozyme, nanobody, mini Q-body

## Abstract

Quenchbodies (Q-bodies), a type of biosensor, are antibodies labeled with a fluorescent dye near the antigen recognition site. In the absence of an antigen, the dye is quenched by tryptophans in the antibody sequence; however, in its presence, the dye is displaced and therefore de-quenched. Although scFv and Fab are mainly used to create Q-bodies, this is the first report where a single-domain heavy chain V_H_ from a semi-synthetic human antibody library formed the basis. To create a proof of concept “mini Q-body”, a human anti-lysozyme single-domain V_H_ antibody C3 was used. Mini Q-bodies were successfully developed using seven dyes. Different responses were observed depending on the dye and linker length; it was concluded that the optimal linker length for the TAMRA dye was C5, and rhodamine 6G was identified as the dye with the largest de-quenching response. Three single-domain antibodies with sequences similar to that of the C3 antibody were chosen, and the results confirmed the applicability of this method in developing mini Q-bodies. In summary, mini Q-bodies are an easy-to-use and time-saving method for detecting proteins.

## 1. Introduction

Biosensors generally exhibit specificity for a single analyte or a group of similar analytes. In recent years, the fluorescence-based detection of molecular targets has attracted major interest [1]. However, many of the available assays are time-consuming and involve laborious steps. When developing novel assays, specific focus should be given to decreasing the handling time and reducing the number of steps involved. Therefore, a need for a more direct approach, such as the use of probes, has presented itself [2].

Owing to the specific binding ability of antibodies, they have gained enormous relevance in the field of biosensors. Traditionally, they were used to detect target molecules, such as specific antigens and toxic compounds, in various samples by enzyme-linked immunosorbent assay (ELISA) [3]. However, conventional ELISA relies on several blocking, binding, and washing steps, which are time-consuming and cumbersome. The demand for a direct approach has led to the development of several antibody-based biosensors. Among them, fluorescent immunosensors called Quenchbodies (Q-bodies) are based on changes in fluorescent properties in response to antibody–antigen interactions. This method has been extensively studied owing to its ease of use, retention of high specificity, and sensitivity [4].

Q-bodies are based on antibody formats that are fluoro-labeled at the N-terminus of the antibody variable region (Fv) of the heavy and/or light chain(s), and function by quenching or de-quenching fluorophores upon binding to their targets. Many biosensors involving fluorophores are based on Förster resonance energy transfer, in which the energy created by the fluorescence excitation of a fluorophore is transferred to an adjacent fluorophore [5]. However, this requires two fluorophores, one to function as an acceptor and the other as a donor, which limits the range of fluorophore pairs to prevent overlapping of their emission spectra. Furthermore, the change in the Förster resonance energy transfer signal is sometimes smaller than the background fluorescence; thus, a greater number of measurements is needed to have sufficient statistical power to distinguish small positive signals from the background noise [6]. Another approach involving labeling with environmentally sensitive fluorophores has recently been developed [7]. This is based on the coupling of solvatochromic fluorophores to residues near the antigen-binding site of an antibody. The main issue with this approach is that the target molecules reported to date are all proteins with large epitopes, thereby limiting the potential of this method. Similarly, the main principle behind the Q-body approach is that certain amino acids—mainly tryptophan (Trp)—are required to quench the dye in the variable region; this approach relies on photo-induced electron transfer. Although biosensors based on peptide or nucleotide backbones that rely on photo-induced electron transfer from Trp or guanidine derivatives have also been reported [8], Q-bodies have the unique characteristic of using antibodies as a scaffold.

The advantage of Q-body technology compared to ELISA-based technologies is its ability to rapidly quantify target compounds without any need for a washing step. Measurements can be carried out by simply mixing the Q-body with an antigen and detecting the fluorescence, thereby reducing time consumption. Additionally, the target compound can be a wide array of molecules, from small molecules, including narcotics such as morphine [4,9,10,11] and small peptides such as BGP-C7 [12], to large proteins such as influenza hemagglutinin and SARS-CoV-2 proteins [13,14,15,16]. Both cell-free and *Escherichia coli*-based methods have been previously applied to produce Q-bodies. The main differences between these methods are the chemical structural variations of the linkers and proteins owing to the different labeling methods used in the two expression systems. In cell-free expression systems, the dye is attached to the antibody by incorporating a non-natural amino acid-linked amber aminoacyl-tRNA during translation, whereas it is attached by chemical labeling through a thiol-maleimide group when expressed in *E. coli*.

Single-domain antibodies (sdAbs) are domain antibodies devoid of light chains and have been discovered in nature within the Camelidae family and in some cartilaginous fishes [17]. This format represents one of the smallest available Ig fragments that has a functional paratope [18]. In recent years, recombinant sdAbs have become popular in the fields of biotechnology and medicine. The small size of sdAbs (approximately 15 kDa) presents a huge advantage, as sdAbs are generally more stable, easier to produce, and penetrate tissues more easily than conventional antibodies [19]. Furthermore, because of their size, sdAbs have been demonstrated to effectively interact with the active sites of enzymes [20], where binding often takes place in cavities. Among them, C3 sdAb was selected against hen egg lysozyme (HEL) using the phage-displayed library Predator [21]. The specific binding mechanism of C3 toward lysozyme has not yet been elucidated; however, it binds lysozyme extremely efficiently. In this study, a novel mini Q-body format based on C3 sdAb is introduced, as all conventional Q-bodies rely on larger antibody fragments.

## 2. Materials and Methods

### 2.1. Materials

KOD-Plus-Neo, T4 DNA polymerase, and Ligation High ver. 2 were obtained from Toyobo (Osaka, Japan). Restriction enzymes and *E. coli* SHuffle T7 Express lysY [22,23] were obtained from New England Biolabs Japan (Tokyo, Japan). Oligonucleotides (Appendix A) were obtained from Operon Eurofins (Tokyo, Japan). The PureYield plasmid miniprep kit was obtained from Promega (Madison, WI, USA). Talon metal affinity resin and a Talon disposable gravity column were obtained from Takara-Bio (Otsu, Japan). The immobilized tris(2-carboxyethyl)phosphine (TCEP) disulfide-reducing gel was obtained from Thermo Pierce (Rockford, IL, USA). ATTO520-C2-mal cells were obtained from ATTO-TEC (Siegen, Germany). Rhodamine6G (R6G)-C5-mal and 5(6)-TAMRA-C0-mal were obtained from Setareh Biotech LLC (Eugene, OR, USA). TAMRA-C2-mal was obtained from Anaspec (Fremont, CA, USA). TAMRA-C5-mal was obtained from Biotium (Hayward, CA, USA). Rhodamine red (Rho)-C2-mal cells were obtained from Life Technologies (Carlsbad, CA, USA). His Mag Sepharose Ni was obtained from GE Healthcare (Piscataway, NJ, USA). Anti-DYKDDDDK antibody magnetic beads and DYKDDDDK peptide were obtained from Fujifilm Wako Pure Chemicals (Osaka, Japan). The Nanosep Centrifugal-3 k Ultrafiltration Device was obtained from Pall Corporation (Ann Arbor, MI, USA). The Costar 3590 microplate was obtained from Corning Japan (Tokyo, Japan). The PentaHis-HRP conjugate was obtained from Qiagen (Hilden, Germany). 3,3′,5,5′-Tetramethylbenzidine (TMBZ) was obtained from Sigma (St. Louis, MO, USA). Unless otherwise indicated, all other chemicals and reagents were obtained from Fujifilm Wako Pure Chemical Industries, Ltd. (Osaka, Japan).

### 2.2. Vector Construction for Expression of sdAb

The pSQ-sdAb vector was generated by removing the scFv region and inserting the C3 gene into pSQ(KTM219) [24]. The C3 gene was amplified using the primer set FWR-Age and REV-BamHI (Appendix A), which introduced an AgeI site at the beginning of the gene and a BamHI site after the His-tag gene at the 3′ end. After amplification of the gene using polymerase chain reaction (PCR), the PCR products were separated on a 2% agarose gel, and the relevant DNA bands were identified using UV light. Following confirmation of insertion amplification, the DNA was digested with the restriction enzymes AgeI and BamHI for 1 h at 37 °C and purified using a Wizard^®^ SV Gel and PCR Clean-Up System (Promega). The pSQ(KTM219) vector was digested using the same restriction enzymes (AgeI and BamHI) to cut out the entire reading frame, except for the Cys-tag (MAQIEVNCSNET) at the N-terminal and the C-terminal FLAG-tag (DYKDDDDK). Next, the digested product was electrophoresed on a 1% agarose gel, and digestion was confirmed by UV light. The target DNA band was excised from the gel and then purified. A mixture of digested vector DNA and insert DNA at a ratio of 3:1 was prepared, and an equal volume of Ligation High ver. 2 was added. The mixture was incubated at 16 °C for 30 min before transformation into XL10-gold competent *E. coli* and plating on an LB agar plate with 1% glucose and 100 μg/mL ampicillin. Random colonies grown on these plates were selected for colony PCR and sequence analysis using the T7 promoter and T7 terminator. The vector with the correct sequence was designated pSQ-C3 that was then used for preparation of immunosensor as shown in Figure 1a. Genes of the C3 mutants W36F, W47F, and W103F were prepared by site-directed mutagenesis following the manufacturer’s protocol, using primer sets W36toFback/W36toFbfor, W47toFback/W47toFfor, and W103toFback/W103toFfor, respectively, before amplification with FWR-Age/REV-BamHI and cloning, as above.

### 2.3. Synthesis and Purification of sdAb Fragments

A single colony of pSQ-C3 transformed SHuffle T7 Express lysY cells was picked and grown at 30 °C and 200 rpm in 4 mL of Luria-Bertani medium containing 100 μg/mL ampicillin until an OD_600_ of 0.9 was achieved. Subsequently, 2 mL was used to inoculate 100 mL of the medium and cultured at 30 °C and 200 rpm until an OD_600_ of 0.5 was attained. Next, protein expression was induced with isopropylthio-β-galactopyranoside at a final concentration of 0.5 mM, and the culture incubated for 16 h at 16 °C and 200 rpm.

The culture was centrifuged at 8000× *g* for 10 min at 4 °C, and the supernatant was removed. The pellet was resuspended in 5 mL of Talon wash buffer (50 mM phosphate, 0.3 M sodium chloride, 5 mM imidazole, pH 7.4), and crushed by sonication (Multi shot model, Constant Systems Ltd. (Daventry, UK)) at 20 kpsi. Next, the sample was centrifuged at 8000× *g* and 4 °C for 10 min, and the supernatant was isolated and mixed with 100 μL of Talon-immobilized cobalt affinity beads (Takara-Bio, Shiga, Japan) and incubated on a rotator wheel at 25 °C for 30 min. After incubation, the protein sample was centrifuged (100× *g*, 1 min, 4 °C), and the supernatant was discarded. The precipitate was then dissolved in 25 mL of Talon buffer and centrifuged (100× *g*, 1 min, 4 °C), followed by discarding the supernatant, dissolving the precipitate in 5 mL of Talon buffer, and adding the sample to an empty Talon affinity column (Takara-Bio). The column was washed three times with 8 mL Talon washing buffer, followed by the addition of 4 mL Talon elution buffer (50 mM phosphate, 0.3 M NaCl, 0.5 M imidazole, pH 7.4). The eluate was analyzed by sodium dodecyl sulfate-polyacrylamide gel electrophoresis (SDS-PAGE).

### 2.4. Generation of Mini Q-Bodies

Protein samples were subjected to buffer exchange to remove imidazole. Nanosep centrifugal-3 k ultrafiltration (Pall Corporation) was performed as described in the manufacturer’s protocol. After changing the buffer to phosphate-buffered saline with 0.5% Tween 20 (PBST), a TCEP bead volume equivalent to 450 μg of purified protein was added and the samples were incubated on a rotator wheel for 1 h at 25 °C. The samples were then centrifuged (1000× *g*, 1 min, 4 °C) and the supernatant was recovered. The supernatant was divided into 50-μg samples and a 20-fold molar excess of one of the following fluorescent dyes was added to each sample: TAMRA-C0-mal, TAMRA-C2-mal, TAMRA-C5-mal, TAMRA-C6(5)-mal, TAMRA-C6(6)-mal, ATTO520-C2-mal, and R6G-C5-mal (Appendix A). The samples were incubated at 4 °C on a rotator wheel for 16 h in the dark. After labeling, the protein was purified using a Flag-tag. In the first step, 20 μL of Anti-DYKDDDDK antibody magnetic beads was added, and the sample was incubated on a rotator wheel at 25 °C in the dark for 1 h. The samples were then subjected to centrifugation (1000× *g*, 1 min, 4 °C) followed by discarding of the supernatant. The precipitate was then washed three times with 1 mL of Flag washing buffer (20 mM phosphate, 0.5 M NaCl, 5 mM imidazole, pH 7.4) and centrifuged (1000× *g*, 1 min, 4 °C). Afterwards, Flag peptide buffer (15 μL of 5 mg/mL 3xFlag peptide and 85 μL of Flag buffer) was added and the samples were incubated on a rotator wheel for 1 h at 25 °C. Finally, the samples were centrifuged (1000× *g*, 1 min, 4 °C) and the supernatant containing the labeled mini Q-body of interest was recovered. The purified mini Q-bodies were analyzed using SDS-PAGE.

### 2.5. ELISA

The HEL antigen (2 μg/mL in 100 μL PBS) or control bovine serum albumin (BSA, 2 μg/mL in 100 μL PBS) was added in triplicate directly to Microlon 655,061 microplate wells (Greiner Bio-one, Kremsmünster, Austria). The plate was incubated for 16 h at 4 °C, followed by the addition of 20% Immunoblock for 2 h at 25 °C. Next, the wells were washed three times with PBST, the target antibody was added at a concentration of 2 μg/mL in a volume of 100 μL, and the samples were incubated for 1 h at 37 °C. Subsequently, the wells were washed three times with PBST and the bound protein was probed with a 5000-fold diluted anti-His-HRP antibody in PBST for 1 h at 25 °C. After incubation, the wells were again washed three times with PBST, developed with 100 μL of substrate solution (100 μg/mL TMBZ, 0.04 μg/mL hydrogen peroxide in 100 mM sodium acetate, pH 6.0), and incubated for 15 min. The reaction was stopped with 100 μL of 1 M hydrochloric acid and the absorbances were read at 450 nm and 655 nm on a microplate reader Model 680 (Bio-Rad, Shanghai, China). PBS without any antigen was used as a control, and the same procedure was performed.

### 2.6. Fluorescence Measurements

Samples were measured using a 5 × 5 mm quartz cell by adding 10 ng of mini Q-bodies in 250 μL of PBST. De-quenching of the samples was performed by the addition of 10 μM antigen (Figure 1b) and measured after 5 min of incubation. Dose–response measurements were performed by the addition of antigen in a titration at 5 min intervals followed by measuring the fluorescence spectra using a Model FP-8500 spectrofluorometer (JASCO, Tokyo, Japan). Regarding samples measured during denaturation (Figure 1b), the protein structure was irreversibly unfolded by adding 250 μL of 7 M guanidium hydrochloride (GdnHCl) with 100 mM dithiothreitol (DTT). The incubation time was increased to 10 min. Both the excitation and emission slit widths were set to 5.0 nm. The excitation wavelengths were 520, 530, and 546 nm for the ATTO520-, R6G-, and TAMRA-labeled mini Q-bodies, respectively. The pertinent dose–response results were fitted using intensity levels at the maximum emission wavelength of each mini Q-body. Additionally, Kaleida Graph 4.1 was used to create these curves and calculate the EC_50_ values by fitting them according to a 4-parameter logistic equation.

## 3. Results

### 3.1. Expression of sdAb C3

After purification by using immobilized metal affinity resin for His-tag, SDS-PAGE was performed to confirm the purity of sdAb. As shown in Figure 2a, the protein showed the expected size of approximately 17 kDa. No large difference was observed between the reduced and non-reduced samples, possibly owing to the small size of the protein. The concentration of the purified sample was determined to be 847 μg/mL after an average concentration value was calculated from the spectrophotometry and Bradford assay results. To test whether the protein was folded and functional, ELISA was performed. The C3 sdAb was tested against both its cognate antigen lysozyme and an unrelated protein, BSA. The results showed specificity toward lysozyme, while the signals were much weaker when there was no protein or BSA present in the wells (Figure 2b).

### 3.2. Labeling and Purification of C3

To determine which fluorescent dye could function optimally in combination with a V_H_ single-domain antibody, a wide range of organic dyes with different spectra and properties were investigated. Additionally, the distance between the N-terminal attachment point (Cys residue) and the fluorescent dye was examined using a range of commercially available TAMRA-maleimide dyes with five different linkers (C0, C2, C5, C6(5), and C6(6)). Ideally, the N-terminal thiol groups should not form a disulfide bond, but it is possible that the Cys-tag binds to another Cys-tag in the neighboring sdAb. As these disulfide bonds are not shielded by the protein structure, it is important to reduce them by adding TCEP beads, allowing protein labeling when the dye is added after removal of the TCEP beads. Next, the protein was purified by the addition of anti-DYKDDDDK beads, followed by extensive washing by centrifugation to remove any free dye. Meticulous washing is the determining factor for the successful reduction of background signals, thus obtaining a high signal-to-background ratio. Finally, the protein was detached from the anti-DYKDDDDK beads by adding Flag peptide, which has a high affinity toward the beads. To examine the level of success regarding labeling and purity, the samples were run on SDS-PAGE and measured under UV light, followed by Coomassie blue staining to determine the loss of protein concentration.

As expected, Coomassie Brilliant Blue staining revealed generally pure protein samples (Figure 3a), even though their concentrations decreased significantly. Figure 3b also shows that there is a strong fluorescent band at approximately 18 kDa in each lane, which indicates that each protein was successfully labeled.

### 3.3. Antigen-Binding Activity of Q-bodies

To confirm the antigen-binding activity of C3 proteins after fluorescence labeling, ELISA was performed on all seven labeled samples. After antigen HEL or non-antigen BSA was immobilized in the wells, labeled or unlabeled C3 was added. After incubation, the amount of bound protein was determined to confirm the specific binding. As shown in Figure 3c, the ELISA results clearly confirmed that the labeled samples retained a strong affinity toward the antigen after labeling. Comparing the signals of labeled and unlabeled C3 sdAbs, no significant differences were observed in any of the seven cases, indicating that the affinity of the antibody was not significantly affected by the addition of these dyes at the Cys-tag. In addition, negligible signals were observed in the absence of antigen.

### 3.4. Fluorescence Changes of Mini Q-Bodies with and without Antigen

The fluorescence intensity of the mini Q-body was measured both with and without the antigen. A significant increase in the fluorescence intensity signal was observed in the presence of a high concentration of antigen (10 μM), and all seven dyes could be quenched and de-quenched with varying de-quenching responses (Appendix A). A significant difference in the baseline and de-quenching properties of the different samples was observed; nonetheless, all seven dyes were able to function properly as mini Q-bodies. Results were obtained by measuring in triplicate with and without antigens. Although all the labeled samples had similar binding activities during ELISA, a prominent difference in the de-quenching ability was observed between the different dyes. Additionally, a simultaneous control experiment was conducted to measure the fluorescence intensity in the presence of BSA. The samples measured with the control antigen showed no change in fluorescence intensity.

### 3.5. Comparison of De-Quenching Efficiency under Denaturation or Antigen Addition

When mini Q-bodies are denatured using chaotropic and reducing agents, the interaction between the fluorescent probe and the Trp residues in the sdAb is relieved, leading to de-quenching. The degrees of de-quenching by antigens and denaturation were compared. Ten nanograms of mini Q-bodies were added to 7 M guanidine HCl with 100 mM DTT, measured after 10 min of incubation, and compared to 10 ng mini Q-bodies de-quenched in the presence of 10 μM antigen. The de-quenching response from denaturation should be the maximal response available, and is therefore extremely important to identify. In general, the changes in fluorescence after denaturation correlated with the changes in fluorescence after the addition of antigen, indicating quenching of the dye by sdAb in its native state and almost complete release after the addition of antigen. As shown in Figure 4, TAMRA-C0 dye was fully de-quenched by both the antigen and denaturation, supporting the theory that the dye was initially only modestly quenched. In contrast, the other dyes (except R6G) did not achieve the same de-quenched level as when the protein was denatured, indicating that the protein was not fully able to de-quench with only the antigen present. Furthermore, a peculiarity was observed for R6G, as the dye achieved higher de-quenching in the presence of antigen than when the protein underwent denaturation. The lower response upon denaturation might indicate that R6G is quenched to some extent by dye–dye interactions when the protein itself is denatured in this condition.

### 3.6. Dose–Response Curve for Antigen Detection

To test if the fluorescence intensity could be controlled by antigen concentration, fluorescence intensity was measured at varying concentrations of antigen: 0, 1, 3, 10, 30, 100, and 300 nM, 1 μM, 3 μM, and 10 μM (Figure 5a). The measurements were performed by adding increasing amounts of antigen to the cuvette, followed by a 5 min incubation period before adding more antigen during titration. In all seven cases, a dose response was obtained, with varying degrees of success. In summary, the TAMRA-C5-labeled mini Q-bodies exhibited the most pronounced dose response among the TAMRA-labeled mini Q-bodies. However, the R6G-C5-labeled mini Q-bodies showed the highest response overall, surpassing all the TAMRA-labeled Q-bodies (Figure 5b). The sensitivity of the assay with R6G-C5 Q-body was evaluated to be around 30 nM of antigen, which matched well with the affinity of C3 [21]. The C3 domain antibody is of modest affinity, 5 μM; by affinity maturation the sensitivity will increase, potentially to the sub-nanomolar. The half maximum efficiency (EC_50_) values of each mini Q-body are summarized in Table 1.

### 3.7. Analysis of Quenching Mechanism

To investigate the role of each conserved Trp residue in the framework region in antigen-binding activity and quenching, three C3 mutants, W36F, W47F, and W103F, were prepared and converted to mini Q-bodies; by labeling with TAMRA-C5, their antigen-binding activity and the degree of quenching was investigated. As shown in Figure 6a, W47F and W103F retained almost the same degree of antigen-binding activity of C3WT while W36F lost, indicating that W36 plays an important role in maintaining antigen-binding properties of C3. After labeling with TAMRA-C5, W47F, and W103F still have antigen-binding activity as strong as TAMRA-labeled C3WT, as shown in Figure 6b. Fluorescence response upon antigen binding of WT, W47F, and W103 was also investigated. As shown in Figure 6c, compared with the wild-type C3, the W47F mutant lost its 3/4 quenching/de-quenching behavior, while W103F lost half, suggesting that these two Trps play major roles in quenching.

### 3.8. Other HEL sdAbs and Their Derived Mini Q-Bodies

Three other sdAbs against HEL—1B2, 4D5, and 4A11—were also obtained from the same phage display library as the C3 antibody ([21] and unpublished) and were converted to mini Q-bodies by labeling with TAMRA-C0, TAMRA-C2, TAMRA-C5, TAMRA-C6(5), TAMRA-C6(6), ATTO520, and R6G. The HEL-binding activity was analyzed by ELISA and, as shown in Appendix A, the unlabeled sdAbs and their derived mini Q-bodies showed similar antigen-binding activity. The de-quenching of each mini Q-body upon antigen binding was also investigated and, as shown in Figure 7, similar patterns like sdAb C3 were obtained.

## 4. Discussion

V_H_-based mini Q-bodies were successfully prepared using an anti-HEL single-domain antibody. Quenching of fluorescent dyes is thought to be achieved by photo-induced electron transfer from Trp residues, as shown for previously designed Q-bodies. From the location of the antigen-binding loop in the X-ray structure of the similar anti-HEL V_H_ H04 [25,26] shown in Figure 8a, W47 is situated at the position nearest to the binding site, while W103 (W111 in Figure 8b) is located nearest in its primary structure. This could satisfactorily explain the possible quenching of the dye conjugated to the N-terminus, and also its de-quenching owing to antigen binding at the CDR3 loop. If the structure and antigen-binding mode of the target V_H_ is similar to these antibodies, we can expect a high probability of success when converting them to Q-bodies. Recently, autonomous knob domain peptides were described, constituting small antigen-binding modules that can be synthesized chemically [27,28]. It would be very interesting to test whether these peptides can be converted to equivalents of the mini Q-bodies.

Regarding the linker length in the mini Q-bodies, TAMRA-C5 had the best quenching/de-quenching phenotype among all the TAMRA derivatives used. However, dyes with longer linkers C6(5) and C6(6) exhibited somewhat reduced quenching/de-quenching. Consequently, it appears that the optimal linker length for TAMRA is C5. Compared with TAMRA, both ATTO520 and R6G achieved significantly higher responses. De-quenching responses of 3.73-fold for ATTO520 and 5.79-fold for R6G were observed compared to TAMRA-C5 (3.01-fold). Although TAMRA-C2 did not function optimally, ATTO520 with a C2 linker still achieved a higher dose response than TAMRA-C5, indicating that the ATTO520 dye might be better suited to creating mini Q-bodies. The linker length is a very important contributor to optimal quenching; therefore, it would be of great interest to examine the results of ATTO520 attached to a C5 linker. However, there are also other factors, such as hydrophobicity and molecular size, that might be instrumental in creating optimal mini Q-bodies.

R6G showed the greatest promise and was attached by a C5 linker, again indicating that a C5 spacer length might be optimal for enhancing the dye’s flexibility without hindering its quenching ability. As no other linkers were tested with ATTO520 or R6G, it is also possible that these two dyes are better suited to creating mini Q-bodies, and investigating the optimal linker lengths for these two dyes might yield improved de-quenching.

## 5. Conclusions

Several mini Q-bodies were successfully constructed based on anti-HEL V_H_. The results demonstrate that it is possible to produce single-domain Q-bodies with a wide array of organic dyes. Surprisingly, compared to a single-chain Fv-based Q-body with an optimized response of ~2.5-fold [29], superior responses were obtained for ATTO520- and R6G-labeled mini Q-bodies, and R6G was identified as the best-suited dye for creating mini Q-bodies owing to its strong de-quenching response. The degree of de-quenching in all cases could be regulated by the amount of antigen present, thus confirming the sensitivity of this assay at a level of 30 nM ligand. Throughout this project, the results obtained from the experiments indicate that Trp in the framework region of the antibodies is essential for their functioning as mini Q-bodies.

## Figures and Tables

**Figure 1 sensors-23-02251-f001:**
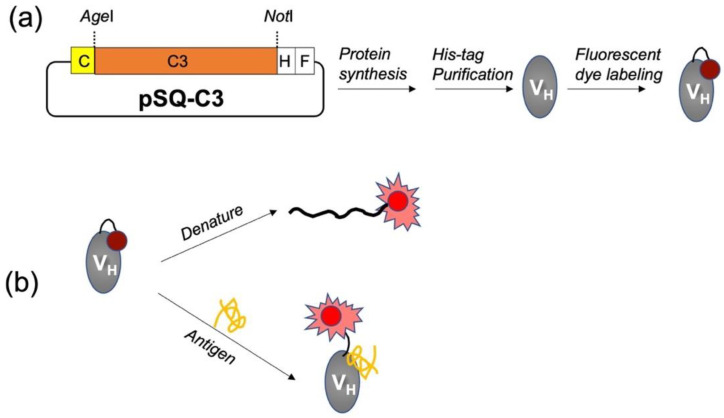
Schematic images of the generation (**a**) and evaluation (**b**) of mini Q-bodies. C, H, and F indicate Cys-tag, His-tag, and FLAG-tag, respectively.

**Figure 2 sensors-23-02251-f002:**
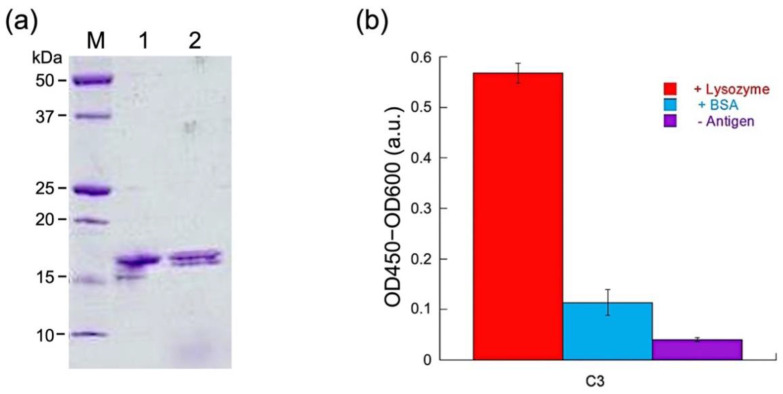
(**a**) SDS-PAGE analysis after Talon purification of C3 protein. M: Marker, lane 1: non-reduced C3, lane 2: reduced C3. (**b**) ELISA test of C3, with/without the antigen lysozyme. BSA: negative control with bovine serum albumin.

**Figure 3 sensors-23-02251-f003:**
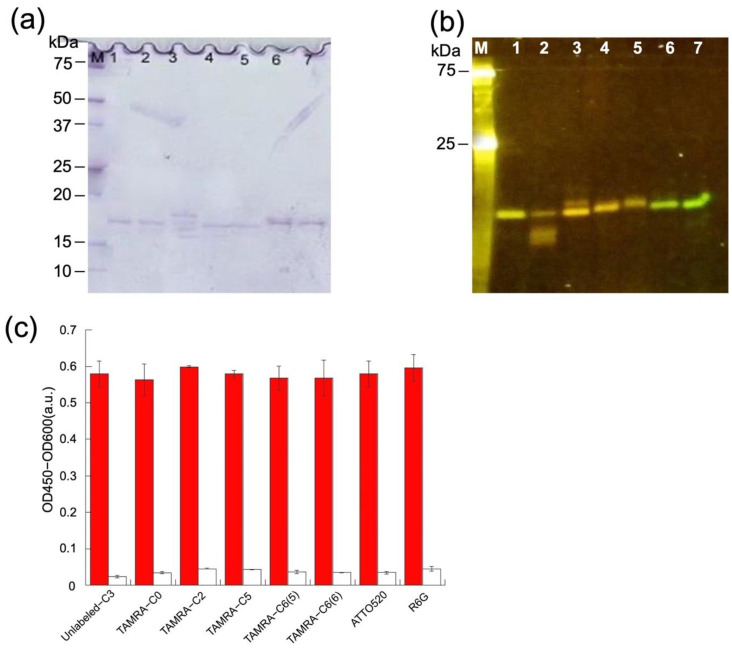
(**a**) Coomassie Brilliant Blue staining and (**b**) fluorescence image of the gel after SDS-PAGE for detecting mini Q-bodies. The estimated molecular mass of C3, including tags, is 20 kDa. Lane 1–7: TAMRA-C0, TAMRA-C2, TAMRA-C5, TAMRA-C6(5), TAMRAC6(6), ATTO520, and R6G. (**c**) Specific binding of the mini Q-bodies and unlabeled C3 protein to its antigen, measured by ELISA. The colored bars indicate the signal measured while the white bars stand for the signal in the absence of antigen. Error bars show the standard deviation (*n* = 3).

**Figure 4 sensors-23-02251-f004:**
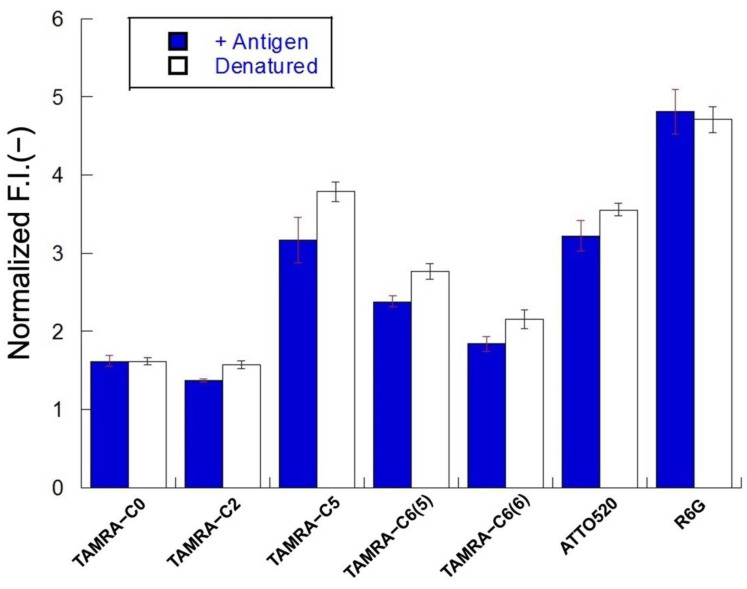
Normalized fluorescence intensities of mini Q-bodies in the presence of 10 μM of antigen and the presence of a denaturant. Error bars represent ±1 SD (*n* = 3).

**Figure 5 sensors-23-02251-f005:**
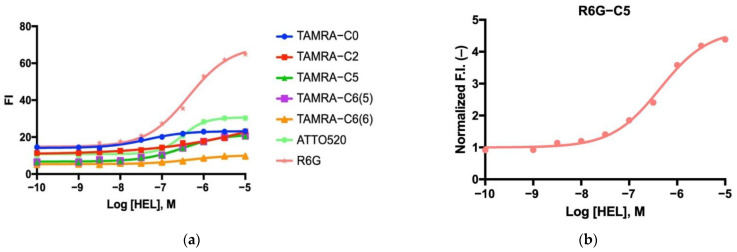
(**a**) Dose–response curve of the mini Q-bodies labeled with different dyes. (**b**) Rhodamine 6G-labeled mini Q-body dose–response curve.

**Figure 6 sensors-23-02251-f006:**
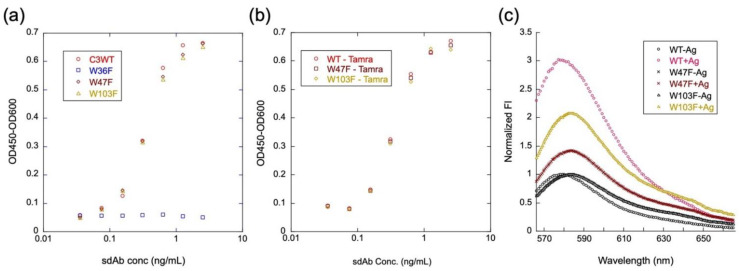
Contribution of each Trp in C3 to the antigen-binding activity and quenching of TAMRA-C5-labeled C3 mini Q-body. ELISA analysis of the antigen-binding activity of C3 and its mutants (**a**) and TAMRA-C5-labeled C3 mini Q-body and its mutants (**b**); de-quenching analysis of TAMRA-C5-labeled C3 mini Q-body and its mutants (**c**).

**Figure 7 sensors-23-02251-f007:**
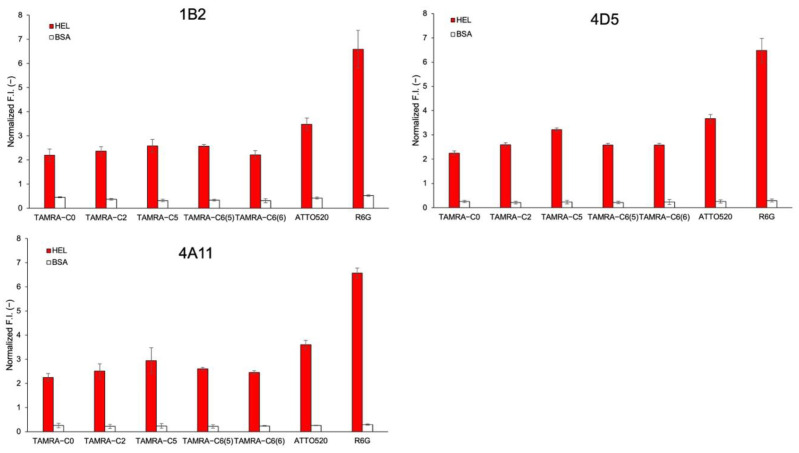
Normalized fluorescence intensities of mini Q-bodies derived from 1B2, 4D5, and 4A11 in the presence of 10 μM of HEL and BSA. Error bars represent ±1 SD (*n* = 3).

**Figure 8 sensors-23-02251-f008:**
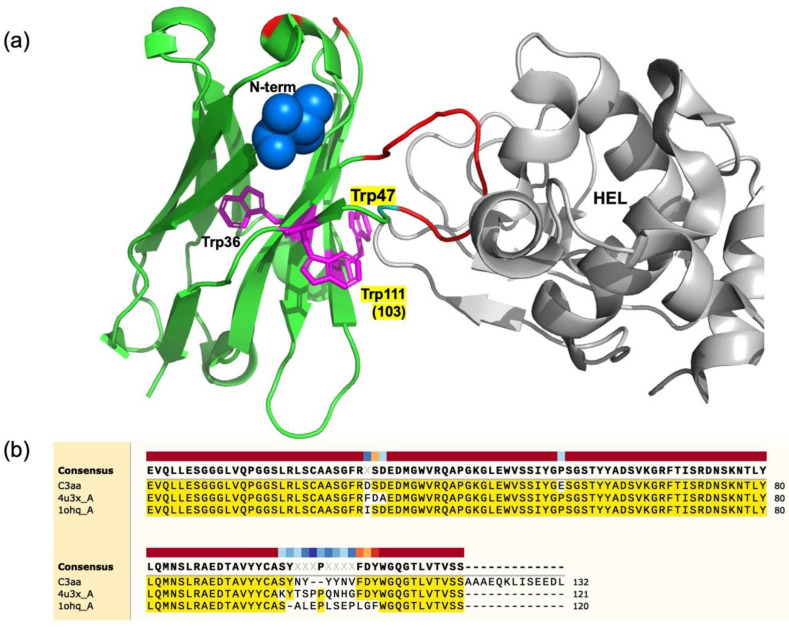
(**a**) Crystal structure of V_H_ H04, an anti-HEL human sdAb complexed with HEL (PDB 4U3X). The location of the V_H_ N-terminus and three Trp residues are shown as a blue ball and magenta sticks, respectively, while amino acid residues different from those of C3 are colored in red, including the long CDR3 loop in the HEL active site. (**b**) Comparison of the primary structures for C3, H04 (4u3x_A), and HEL4 (1ohq_A), which is a stable V_H_ and also the mother V_H_ before phages display selection of C3.

**Table 1 sensors-23-02251-t001:** EC_50_ values of mini Q-bodies prepared with different dyes.

Fluorescent Dye Name	EC_50_ (μM)
TAMRA-C0	0.039
TAMRA-C2	0.050
TAMRA-C5	0.263
TAMRA-C6(5)	0.401
TAMRA-C6(6)	0.362
ATTO520	0.276
R6G	0.397

## Data Availability

The data supporting the results and findings of this study are available within the paper and the Appendix A.

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
