# Peer review of "VH-Based Mini Q-Body: A Novel Quench-Based Immunosensor"

_sensors, 2023, doi:10.3390/s23042251_

Round 1
Reviewer 1 Report
The manuscript describes internally quenched fluorescent VHH domains, which fluoresce on antigen binding. The concept is neat and of relevance to Sensors, and I enjoyed reading the text.
The C3 VHH is only of modest affinity (5 micromolar in reference 21), so that the sensitivity of the Q-Body is limited to around 30 nM for lysozyme. It would be interesting to see how sensitivity could be improved with a much higher affinity antibody domain. Perhaps the authors could comment on the relationship between affinity and sensitivity.
The denaturation experiment is a good positive control and shows how de-quenched the tag is on antigen binding.
On line 83 it is claimed that the VHH is the smallest available Ig fragment that has a functional paratope. With the recent invention of autonomous knob domain peptides Isolation of antigen-specific, disulphide-rich knob domain peptides from bovine antibodies (plos.org)
(Macpherson PLOS Biology 2020) this is no longer the case. As these knob domains can be chemically synthesised The Chemical Synthesis of Knob Domain Antibody Fragments (acs.org)
(Macpherson ACS Chemical Biology 2021) it may be possible to incorporate the fluorescent tag during the synthesis and in a precise position in relation to a relevant tryptophan residue. It might be worth the authors acknowledging and considering these constructs in future work.
A few minor points:
line 348 figure 6 legend needs a (c).
line 352 converted
Author Response
We would like to thank both reviewers for constructive and positive feedback. We agree on the suggestions for minor revisions raised by the reviewers and have addressed them as follow:
Reviewer1
Comment:
The manuscript describes internally quenched fluorescent VHH domains, which fluoresce on antigen binding. The concept is neat and of relevance to Sensors, and I enjoyed reading the text.
The C3 VHH is only of modest affinity (5 micromolar in reference 21), so that the sensitivity of the Q-Body is limited to around 30 nM for lysozyme. It would be interesting to see how sensitivity could be improved with a much higher affinity antibody domain. Perhaps the authors could comment on the relationship between affinity and sensitivity.
Response:
Thank you for raising this issue. We have added comment on the relationship between affinity and sensitivity as follows (line 323-325):
‘The sensitivity of the assay with R6G-C5 Q-body was evaluated to be around 30 nM of antigen, which matched well with the affinity of C3[21]. The C3 domain antibody is of modest affinity, 5uM, by affinity maturation the sensitivity will increase, potentially to the sub-nanomolar’
Comment:
The denaturation experiment is a good positive control and shows how de-quenched the tag is on antigen binding.
On line 83 it is claimed that the VHH is the smallest available Ig fragment that has a functional paratope. With the recent invention of autonomous knob domain peptides Isolation of antigen-specific, disulphide-rich knob domain peptides from bovine antibodies (plos.org)
(Macpherson PLOS Biology 2020) this is no longer the case. As these knob domains can be chemically synthesised The Chemical Synthesis of Knob Domain Antibody Fragments (acs.org)
(Macpherson ACS Chemical Biology 2021) it may be possible to incorporate the fluorescent tag during the synthesis and in a precise position in relation to a relevant tryptophan residue. It might be worth the authors acknowledging and considering these constructs in future work.
Response:
Thank you for raising this issue.
We have modified the sentence (line 83-84) ‘This format represents the smallest available Ig fragment that has a functional paratope’ to ‘This format represents one of the smallest available Ig fragment that has a functional paratope’.
We also added the above two papers to reference list. The following statements have been added to discussion (line 385-388).
‘Recently, autonomous knob domain peptides was described constituting small antigen-binding modules that can be synthesized chemically [28,29]. It would be very interesting to test whether these peptides can be converted to equivalents of the mini-Q-bodies.’
A few minor points:
line 348 figure 6 legend needs a (c).
Response: We have added ‘(c)’ to Figure 6 legend.
line 352 converted
Response: The spelling mistake have been corrected. Thank you very much.

Reviewer 2 Report
This study demonstrated the development of a nanobody and quench-based immunosensor. Overall, the experimental design and results are clear and interesting to readers. The manuscript should be accepted for publication in the journal if the minor revision is well addressed. Below are some specific comments:
1. In line 18, “no attempts have been made to create Q-bodies using monodomains derived from heavy chain VH” is not true, I found one nanobody-based Q-body published before (PMID: 33169966)
2. In line 351, there are several other nanobodies listed, citations might be needed to show the sources.
Author Response
We would like to thank both reviewers for constructive and positive feedback. We agree on the suggestions for minor revisions raised by the reviewers and have addressed them as follow:
Reviewer 2
Comment:
This study demonstrated the development of a nanobody and quench-based immunosensor. Overall, the experimental design and results are clear and interesting to readers. The manuscript should be accepted for publication in the journal if the minor revision is well addressed. Below are some specific comments:
- In line 18, “no attempts have been made to create Q-bodies using monodomains derived from heavy chain VH” is not true, I found one nanobody-based Q-body published before (PMID: 33169966)
Response: Thank you for raising this question. Paper 33169966 reported a Q-body based on nanobody derived from a immunized Llama. However, the mini Q-body reported in this paper was based on a human synthetic Vh domain antibody. Although the molecular weights of them are similar, they are of different origin. We have modified the text to clarify the issue (line 18-19).
´this is the first report where a single domain heavy chain VH from a semi-synthetic human antibody library formed the basis.``
- In line 351, there are several other nanobodies listed, citations might be needed to show the sources.
Response: The antibody fragments mentioned in line 351 was obtained from the same phage antibody library as the C3 binder, thus constituting an antibody fragments with similar scaffold, but different paratope. The text has be modified (line 356-357):
“1B2, 4D5, and 4A11 were also obtained from the same phage display library as the C3 antibody ([21] and unpublished)”
